P-curve accurately rejects evidence for homeopathic ultramolecular dilutions

Reisman Samuel samuel.reisman@downstate.edu
Balboul Mostafa
Jones Tashzna
SUNY Downstate College of Medicine , Brooklyn , NY , United States of America
Stern David
Electronic publication date: 2019 Jan 23
Publication date: 2019
Volume: 7
Electronic Location ID: e6318
Received 2018 Oct 12; Accepted 2018 Dec 16
Copyright: ©2019 Reisman et al.
Copyright year: 2019
Copyright holder: Reisman et al.
License: This is an open access article distributed under the terms of the Creative Commons Attribution License, which permits unrestricted use, distribution, reproduction and adaptation in any medium and for any purpose provided that it is properly attributed. For attribution, the original author(s), title, publication source (PeerJ) and either DOI or URL of the article must be cited.
License URL: https://creativecommons.org/licenses/by/4.0/

Keywords: p-curve, p-value, Publication bias, Ultramolecular dilutions, Homeopathy, Statistics, Evidential value, Statistical significance

Funding: The authors received no funding for this work.

==============================
Background

P-curve has been proposed as a statistical test of evidential value. The distributions of sets of statistically significant p-values are tested for skewness. P-curves of true effects are right-skewed, with greater density at lower p-values than higher p-values. Analyses of null effects result in a flat or left-skewed distribution. The accuracy of p-curve has not been tested using published research analyses of a null effect. We examined whether p-curve accurately rejects a set of significant p-values obtained for a nonexistent effect.

Methods

Homeopathic ultramolecular dilutions are medicinal preparations with active substances diluted beyond Avogadro’s number. Such dilute mixtures are unlikely to contain a single molecule of an active substance. We tested whether p-curve accurately rejects the evidential value of significant results obtained in placebo-controlled clinical trials of homeopathic ultramolecular dilutions.

Results

P-curve accurately rejected the evidential value of significant results obtained in placebo-controlled clinical trials of ultramolecular dilutions. Robustness testing using alternate p-values yielded similar results.

Conclusion

Our results suggest that p-curve can accurately detect when sets of statistically significant results lack evidential value.

Introduction

Most published research papers contain significant p-values (Cristea & Ioannidis, 2018). A significant p-value indicates that obtaining the result within the null distribution is improbable. For example, a p-value of .05 (our assumed significance level throughout this article) is obtained in a null distribution, on average, once every twenty tests. The unlikeliness of obtaining a significant result when effects do not exist is what makes significant p-values significant.

Significant p-values can be misleading. Researchers can increase their odds of obtaining a statistically significant result by performing many studies (Rosenthal, 1979) or many analyses (Ioannidis, 2008). If five independent statistical tests are performed, the likelihood of obtaining at least one significant p-value approaches 25%. Significant findings are reported, while analyses that do not obtain significance are often not reported. Significant p-values may thus reflect selective reporting rather than true effects.

Simonsohn, Nelson & Simmons (2014a) introduced p-curve as a method of evaluating sets of statistically significant results for evidential value. P-curve evaluates the distribution of p-values obtained beyond the .05 significance threshold. The distinct distribution of p-values obtained for true effects forms the basis of p-curve. True effects with significant noncentral distributions will have a greater density of low p-values than high p-values (Westfall, Stanley Young & Young, 1993; Hung et al., 1997), resulting in a right-skewed p-curve. P-values obtained for null effects, or type I errors, result in a uniform distribution, since there is an equal probability of obtaining any single p-value within the null distribution. Left-skewed distributions result when statistical analyses correlate with previous, non-statistically significant analyses.

P-curve consists of two sequential statistical tests. First, the p-value distribution is tested for right-skewness (“right skew test”) against the null hypothesis of a uniform distribution. A right skew indicates evidential value. When the initial test fails to reject a uniform distribution, a subsequent test determines whether the skew is less right-skewed (“flatness test”) than would be expected if the studies were powered at 33%. Rejection of even a minimally significant effect indicates that the effect does not exist, is too small to be of practical significance, or is too small to be analyzed at the sample size used by the investigators (Steidl, Hayes & Schauber, 1997). Rejection of at least 33% power indicates that the set of significant p-values lack evidential value.

Analysis of p-curve using simulated statistical results demonstrate a high sensitivity and specificity for detecting evidential value (Simonsohn, Nelson & Simmons, 2014a) but these estimates may not be accurate when analyzing published research. Testing p-curve’s accuracy using published research is important because p-curve is partially blind. There are several unsound methods of obtaining statistical significance that p-curve cannot detect (Simonsohn, Simmons & Nelson, 2015). For example, p-curve cannot detect when data is changed to obtain highly significant results. Even a single fake, highly significant result among many alpha-errors can significantly increase the probability of a right-skewed p-curve (Simonsohn, Simmons & Nelson, 2015). In addition, determined researchers can perform countless statistical tests until a highly significant result is obtained. The prevalence of such practices has not been determined.

The accuracy of p-curve can be assessed when there is independent suspicion or evidence that a set of significant published results lack evidential value. For example, randomized experimental studies that only report results with covariates presumably omit simpler analysis because they did not obtain significance. P-curve of a set of such findings in an academic journal indicated that they lack evidential value, as expected (Simonsohn, Nelson & Simmons, 2014a). Similar support for p-curve’s accuracy can be derived from analyses of replication studies (Simonsohn, Nelson & Simmons, 2014b) and power posing studies (Simmons & Simonsohn, 2017). However, the a priori evidence that these results lack evidential value is based on the methods used, results reported, or replication results, not on the hypothesis tested. P-curve can be more conclusively tested if the status of an effect is known independent of statistical tests.

Placebo-controlled trials of homeopathic ultramolecular dilutions are, by definition, trials of null effects. Homeopathic medicine was founded in the late 18th century by the German physician Samuel Hahnemann on the principle that “let like be cured by like” (Hahnemann, 2001). A central homeopathic principle is the principle of the minimum dose, where medicines are potentiated by serial dilution until no active substance can be detected in solution (Chase, 2017). Ultramolecular dilutions are substances dilutions beyond Avogadro’s number that are unlikely to retain a single molecule of active ingredient (Fisher, 2012). Because ultramolecular dilutions do not contain an active ingredient, they do not differ from a placebo. Clinical trials investigating whether ultramolecular dilutions differ from placebo are comparing two identical substances, and significant p-values obtained in such investigations lack evidential value. We tested whether p-curve accurately rejects the evidential value of a set of significant p-values obtained in placebo-controlled clinical trials of homeopathic ultramolecular dilutions.

Materials & Methods

We searched for peer-reviewed, placebo-controlled clinical trials contained within CORE-hom, a homeopathic clinical trial database of the Homeopathy Research Institute (https://www.hri-research.org/resources/research-databases/core-hom/). We set the database’s control search filter to “placebo” and peer-review filter to “yes” while leaving the search bar empty. By default, studies from 1941 until the present are included in CORE-hom search results.

Inclusion criteria include:

1. Study is accessible to the authors.

2. Study is a clinical trial comparing ultramolecular dilutions to placebo.

3. Study is randomized, with randomization method specified.

4. Study is double-blinded.

5. Study design and methodology result in interpretable findings (e.g., an appropriate statistical test is used).

6. Study reports a test statistic for the hypothesis of interest.

7. Study reports a discrete p-value or a test statistic from which a p-value can be derived.

8. Study reports a p-value independent of other p-values in p-curve.

The first 20 studies, in the order of search output, that met the inclusion criteria were used for analysis.1 When 20 studies powered at just 50% are included in p-curve, the analysis is “virtually guaranteed to find evidential value” (Simonsohn, Nelson & Simmons, 2014a). In the event the 20 results did not meet our inclusion criteria, randomized trials without specified randomization methods were included. If more than one p-value is reported, the p-value most closely representing the original hypothesis was used, and other related p-values were used for robustness testing. If one p-values could not be designated as most closely representing the original hypothesis, the first p-value reported was used, and subsequent p-values used for robustness testing. A p-curve disclosure table can be found on Open Science Framework (http://osf.io/f39be). Unless otherwise indicated, the methods and inclusion criteria were decided upon before commencing data collection and analysis. The methods were preregistered on Open Science Framework (https://osf.io/v2e7g).

P-curve analysis was performed using version 4.06 of the p-curve app, created by Simonsohn, Nelson & Simmons. The app can be accessed here: http://www.p-curve.com/app4/, and its R code here: http://p-curve.com/app4/pcurve_app4.06.r. For more information regarding the statistical modeling and logical framework of the app, see Simonsohn, Simmons & Nelson (2015).

Results

Search results from the CORE-hom database yielded 170 studies. Full-text review yielded 19 studies that met our inclusion criteria (Fig. 1). No studies were excluded solely for not specifying the randomization method. The full output of search results and exclusion rational can be found in our Open Science Framework (OSF) repository at https://osf.io/f39be/. The full output of all p-curve analyses, including a cumulative p-curve, can also be found in the repository.

Figure 1 Flow sheet summarizing the study search and selection with exclusion rationale.

P-curve analysis was performed on the 19 studies that met our inclusion criteria. The resulting p-curve is shown in (Fig. 2).

Figure 2 P-curve of the 19 studies that met inclusion criteria.

The red dotted line represents the expected p-value distribution when no effect exists. The dashed green line represents the expected distribution when an analysis has 33% true power. The full curve is significantly flatter than the distribution expected with 33% true power (p = .037), indicating a lack of evidential value.

The skew test for evidential value analyzed whether the p-value distribution is skewed-right, indicating evidential value. If either the half p-curve (only p-values less than .025) is right-skewed at the .05 significance level, or both the half and full p-curve are right-skewed at the .10 significance level, the test results indicate evidential value (Simonsohn, Simmons & Nelson, 2015). Neither of the criteria is met in our analysis. The half curve is not right-skewed at the p < .05 significance level (p = .06) and the full curve is not significant at the p < .10 significance level (p = .22), thus the p-curve does not indicate evidential value.

The flatness test for inadequate or absent value analyzed whether the p-curve is flatter than an expected curve if the study had 33% true power. If either the full p-curve is flatter than expected at the .05 significance level, or both the half p-curve and a binomial 33% power test are flatter than expected at the .10 significance level, the test results indicate inadequate or absent evidential value. Here the full p-curve is significantly flatter than would be expected with at least 33% power (p = .038), indicating inadequate or absent evidential value.

In summary, p-curve analysis accurately rejects the evidential value of statistically significant results from placebo-controlled, homeopathic ultramolecular dilution trials. This result indicates that replications of the trials are not expected to replicate a statistically significant result. A subsequent p-curve analysis was performed using the second significant p-value listed in the studies, if a second p-value was reported, to examine the robustness of initial results. P-curve rejects evidential value with greater statistical significance (p < .002) (Fig. 3).

Figure 3 p-curve of robustness test using alternate p-values.

The red dotted line represents the expected p-value distribution when no effect exists. The dashed green line represents the expected distribution when an analysis has 33% true power. The full curve is significantly flatter than the distribution expected with 33% true power (p = .0002), indicated a lack of evidential value.

Notably, the study by Frass et al. (2005), which obtained the lowest p-value among the included studies, is flawed by randomization failure. Individuals with COPD were randomized to treatment and placebo groups. The treatment group (n = 25) included five who require long-term oxygen therapy, while the placebo group (n = 25) included nine individuals who require the same treatment. Long-term oxygen therapy was indicated at the time for moderate to severe hypoxemia at rest, or desaturation with activity or at night (Stoller et al., 2010). The higher percentage of oxygen-requiring participants likely indicates more advanced disease among the placebo group than the treatment group. In addition, the reported COPD stages may be inaccurate. The mean COPD stage, ranging from 1-3, is reported as 1.20 for the placebo group. Mathematically, even when assuming all those not on home oxygen had stage 1 COPD, and all those on home oxygen had stage 2 COPD, at least four individuals with stage 1 COPD must have required home oxygen therapy. This seems unlikely, as stage 1 COPD was defined at the time as mild airflow obstruction of which the patient may or may not be aware (Pauwels et al., 2001). When this study is excluded, p-curve rejects evidential value for the full p-curve with greater statistical significance (p = .0002) (Fig. 4).

Figure 4 P-curve results when Frass et al. (2005) is excluded from the analysis.

The red dotted line represents the expected p-value distribution when no effect exists. The dashed green line represents the expected distribution when an analysis has 33% true power. The full curve is significantly flatter than the distribution expected with 33% true power (p = .0086), indicated a lack of evidential value.

Discussion

P-curve analysis accurately rejects evidential value for a set of significant p-values obtained in placebo-controlled clinical trials of homeopathic ultramolecular dilutions. Our results support the ability of p-curve to detect when sets of significant results lack evidential value. A more precise explanation of our results requires a separate analysis for the two sequential tests that comprise the p-curve:

(1) The right skew test analyzes the hypothesis that a set of significant values contain evidential value against the null hypothesis that they do not indicate evidential value. The sensitivity of the right skew test is reflected in the percentage of true effects found to be right-skewed, and the specificity is reflected in the percentage of null effect not found to be right-skewed. Our results support the specificity of the right skew test, as findings without evidential value were accurately rejected.

(2) The flatness test analyzes the hypothesis that a set of significant values has inadequate or absent evidential value. The sensitivity of the flatness test is reflected in the percentage of null effects found to be flat, and the specificity is reflected in the percentage true effects for which the null of skewness is not rejected. Our results support the sensitivity of the flatness test, as findings without evidential value were accurately accepted as flat.

Our findings support p-curve’s ability to accurately reject evidential value but do not establish p-curve’s ability to accurately accept evidential value. A set of findings for an effect known to exist can test whether p-curve accurately accepts the existent effect. Research conducted on effects initially questioned, but later established beyond doubt, may yield analyzable studies.

We compared the results of p-curve to the results of a trim and fill analysis. P-values and sample sizes were used to compute point estimates and confidence intervals for the standard differences in means using a fixed-effect model. The point estimate of the standard difference in means was 0.445. A trim and fill analysis decreased the estimated effect size to 0.343 (Fig. 5) but the result remains highly significant. A classic fail-safe analysis indicated 505 missing, non-significant studies wound bring the p-value higher than the alpha term. Only p-curve accurately rejects the evidential value of the studies.

Figure 5 Funnel plot with trim and fill analysis.

The white dots depict the included studies while the black dots depict the nine missing studies identified by the trim and fill analysis.

A significant weakness of our analysis is reproducibility. A test of p-curve involves three steps. First, a statistical analysis is designated for which the correct answer is a priori known. Second, a literature review is performed to find a set of significant findings for the statistical analysis designated in the first step. Finally, p-curve analysis is performed to determine whether the correct answer is obtained. However, research studies are typically performed to clarify a question. Researchers are unlikely to test hypotheses for which the answers are known. Homeopathic ultramolecular dilutions are a unique example of studies for which the non-existence of effects can be claimed independent of study results. It is unknown whether there are other, similar examples in the literature. Perhaps another set of placebo-controlled clinical trials of homeopathic ultramolecular dilutions can be found and used to reproduce the current analysis.

Another weakness of our analysis is the assumed impotence of homeopathic ultramolecular dilutions. Many proposed mechanisms of action for ultramolecular dilutions have been published, including but not limited to molecular clustering (Samal & Geckeler, 2001), oscillatory effects (Hyland & Lewith, 2002), non-local quantum effects (Weingartner, 2005), and nanobubble-induced superstructures (Demangeat, 2018). However, proposed mechanisms of action for such dilutions are either implausible within the framework of currently known physical laws or would require fundamental revisions to basic sciences such as Biochemistry or Physics (Sehon & Stanley, 2010; Grimes, 2012). Presently, the claimed efficacy of ultramolecular dilutions is discordant with fundamental scientific concepts (National Center for Complementary and Integrative Health, 2018).

Only homeopathy studies were included in the current analysis. There may be heterogeneity in statistical methods among academic fields, limiting the external validity of our analysis. In addition, homeopathy studies may be more likely to have fundamental methodological flaws than other studies. The quality of homeopathic clinical trials is often low (Mathie et al., 2017). We have found several significant flaws in the papers included in our study. Naudé, Stephanie Couchman & Maharaj (2010) perform a Kruskal–Wallis analysis of hours slept, but do not indicate that they ranked the sleep data of the participant. In addition, their second figure indicates values ranging from 32–45 as the total hours of sleep for the treatment and placebo groups, but neither indicate if they are reporting the mean or median hours nor how the reported values relate to the Kruskal–Wallis test indicated in the footnote. Frass, Linkesch & Banyai (2005) report a Kruskal–Wallis analysis of a between-groups categorical variable in Table 3; however, valid tests of proportion result in a value similar to their result. Tveiten & Bruset (2003) do not report the means or a measure of variance for the treatment and placebo groups. In addition, the methods section refers to the Wilcoxon rank-sum test as an analysis of the mean difference between groups. They may have intended to refer to the mean rank difference rather than the mean difference. Reproduction of our findings in other research fields would significantly strengthen our conclusions, but studies of null effects may not exist in other fields.

Given the inherent difficulties noted in testing p-curve from the published literature, analysis of p-curve’s accuracy remains most suited to the domain of statistical simulations. It is important, however, to confirm that simulations match reality, particularly when human behavior is simulated. Our real-world analysis closely matches simulated p-curve results, supporting both the accuracy of p-curve simulations and the accuracy of p-curve results.

Conclusions

P-curve accurately detects the lack of evidential value in a set of significant results obtained for homeopathic ultramolecular dilutions. Our results suggest that p-curve accurately detects when sets of significant results lack evidential value. Reproductions of our analysis in other research fields and with effects known to exist are important but present significant conceptual difficulties.

Additional Information and Declarations

Competing Interests

Author Contributions

Data Availability

1 Although we designated a sample size of 20, there were only a total of 19 studies that met our inclusion criteria among the CORE-hom search results, making the predesignated sample size superfluous.

The authors declare there are no competing interests.

Samuel Reisman conceived and designed the experiments, performed the experiments, analyzed the data, prepared figures and/or tables, authored or reviewed drafts of the paper, approved the final draft.

Mostafa Balboul analyzed the data, approved the final draft.

Tashzna Jones prepared figures and/or tables, authored or reviewed drafts of the paper, approved the final draft.

The following information was supplied regarding data availability:

Raw data for Z scores for main p-curve analysis, Z scores for robustness testing, and Z scores for p-curve excluding Frass can be found at: Reisman, S., Balboul, M., & Jones, T. (2018, November 26). P-curve accurately rejects evidence for homeopathic ultramolecular dilutions. Retrieved from https://osf.io/f39be/.

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
