# Peer review of "P-curve accurately rejects evidence for homeopathic ultramolecular dilutions"

_PeerJ, doi:10.7717/peerj.6318_

## Round 0.1 · original submission · Major Revisions

Please address each of the reviewers comments in detail in a response letter to accompany the re-submission explaining where you have made changes and the reasons for not making changes if that seems appropriate to you. Reviewer 2 sent me a separate email after submitting his review saying that he had changed his mind about the need to do a PET-PEESE analysis. So, you can ignore his comment about doing PET-PEESE. Otherwise, please address all comments.

Reviewer 1 ·

Basic reporting

Please see attachment.

Experimental design

Please see attachment.

Validity of the findings

Please see attachment.

Additional comments

Please see attachment.

Annotated reviews are not available for download in order to protect the identity of reviewers who chose to remain anonymous.

·

Basic reporting

no comment

Experimental design

no comment

Validity of the findings

no comment

Additional comments

Summary,: the paper applies p-curve analysis to the first 20 homeopathy randomized-trials papers found in a search, concluding they lack evidential value.

I am not familiar with homeopathic research thus I will focus on the p-curve analysis aspect of the paper. Overall the paper seems sound, I have relatively minor comments only. My main comment, the only one involving more than minimal work, is #3 below: I strongly recommend the authors analyze the data also using more traditional, pre p-curve, meta-analytical tools.

Comments.
1) The selection of studies is pretty clear up to the point of selecting the first 20 reported by the output of the search, this unfortunately makes the study selection difficult to reproduce. If tomorrow the Homepathy research institute changes their search algorithm, or even adds new studies, the study selection is not reproducible. It's difficult to fix at this point, but as a remedy perhaps the authors can conduct the search again and save the entirety of the output as a text file/spreadsheet, and they indicate which studies they selected. This would allow a skeptic, say, to second guess their study selection decisions relatively easily.

2) Clarify the contribution
The article reads "P-curve remains untested. The sensitivity and specificity of p-curve have been estimated by simulation (Simonsohn, Nelson & Simmons, 2014b) but has not been tested using published research"

P-curve analysis has been applied to multiople published literatures. For example, the paper introducing it runs it on 20 published studies in the Journal of Personality and Social Psychology, selected as likely to be p-hacked vs not. Our follow-up paper applies it to the choice overload literature. In another paper we apply to the power-posing literature.

I am not asking those other papers be cited, but the novelty of this paper be more clearly indicated.
I do think this is the first time a literature which scientists believe for theoretical reasons must be examining a null finding concludes that indeed lacks evidential value using p-curve. It is in a sense a stronger test than that in our JPSP demonstration for the papers are not being selected for seeming suspicious based on the methods used, but rather, based on the hypothesis tested. That's valuable, and different, and worth clarifying.

3) Compare p-curve to other methods
I would strongly urge the authors to conduct a traditional meta-analysis of this literature, and rely on traditional tools for correcting for publication bias: Trim-and-fill and possibly PET-PEESE and maybe Fail-Safe method. This would provide valuable information regarding not only the absolute but also the relative advantage of relying on p-curve to assess evidential value. I assume these other methods will not correct well at all (they will conclude homopathy is super true) but I am interested in seeing those results published regardless (of course). This is actually not a lot of work, with the Z values ready, you are just 2 lines of codes from having the trim-and-fill results. I am thinking of the paper either stating "Only p-curve correctly identifies homepathy lacks evidential value" or "Also p-curve correctly identifies homeopathy lacks evidential value"

4) Clarify p-value extraction vs test
Looking at the p-curve disclosure table I see that at least sometimes the authors selected a p-value from a paper, rather than a test-statistic, and then submitted the Z-value that is associated with it to the p-curve analysis. That's generally OK, but it should be noted, and also make notes when there is ambiguity surrounding that p-value. Specifically, I opened the first paper in the p-curve disclosure table, Chakraborty et al. (2013), and see that they claim to conduct a Bonferroni correction, but it is not clear exactly what they do (how many hypothesis they correct for) and whether the p-values reported have been corrected already or not. Looking at their Table 2 it seems the reported p-values are not corrected (as they are all accompanied by the note that the p-value is not significant using the Bonferroni correction), but that's a baseline comparison, in Table 3 it is ambiguous whether the p-values have been corrected. A footnote discussing this and other such ambiguities, or perhaps a supplement/appendix may be useful; it should be clear to readers which issue existed and how they were dealt with.

Note that a right-skewed distribution of p-values will remain right-skewed after a Bonferroni correction, as it divides (all) p-value by a constant.

5) the p-curve figures are high resolution on the app, but appear blurry here. Maybe the authors did a print-screen, instead they can save as to keep it high-resolution.

6) the app reports a cumulative p-curve as it excludes more and more p-value, since the authors exclude one extreme value they may be interested in it, but maybe not.

7) In the p-curve disclosure table i would crate a column with test results ready to be copy-pasted to the online app, so just the Z=xx.x, wirhtout the p-value before it and the z in ()

---

## Round 0.2 · accepted · Accept

The reviewers are satisfied with the revisions.

# ·

Basic reporting

2nd round, n/a

Experimental design

2nd round, n/a

Validity of the findings

2nd round, n/a

Additional comments

I am satisfied with the revisions